# Leaf Pigments, Surface Wax and Spectral Vegetation Indices for Heat Stress Resistance in Pea

Endale Geta Tafesse [1] , Thomas D. Warkentin [1] , Steve Shirtliffe [1], Scott Noble [2] and Rosalind Bueckert [1,*]

1    Department of Plant Sciences, College of Agriculture and Bio-Resources, University of Saskatchewan, Saskatoon, SK S7N 5A8, Canada; endale.tafesse@usask.ca (E.G.T.); tom.warkentin@usask.ca (T.D.W.); steve.shirtliffe@usask.ca (S.S.)
2    Department of Mechanical Engineering, College of Engineering, University of Saskatchewan, Saskatoon, SK S7N 5A9, Canada; scott.noble@usask.ca
*    Correspondence: rosalind.bueckert@usask.ca; Tel.: +1-306-966-8826

**Abstract:** Pea is a grain legume crop commonly grown in semi-arid temperate regions. Pea is susceptible to heat stress that affects development and reduces yield. Leaf pigments and surface wax in a crop canopy make the primary interaction with the environment and can impact plant response to environmental stress. Vegetation indices can be used to indirectly assess canopy performance in regard to pigment, biomass, and water content to indicate overall plant stress. Our objectives were to investigate the contribution of leaf pigments and surface wax to heat avoidance in pea canopies, and their associations with spectral vegetation indices. Canopies represented by 24 pea cultivars varying in leaf traits were tested in field trials across six environments with three stress levels in western Canada. Compared with the control non-stress environments, heat stress reduced leaf lamina and petiole chlorophyll a, chlorophyll b, and carotenoid concentrations by 18–35%, and increased leaf lamina chlorophyll a/b ratio, anthocyanin and wax concentrations by 24–28%. Generally, greater leaf pigment and wax concentrations were associated with cooler canopy temperature and high heat tolerance index (HTI) values. Upright cultivars had higher HTI values, whereas the lowest HTI was associated with normal leafed vining cultivars. Vegetation indices, including photochemical reflectance index (PRI), green normalized vegetation index (GNDVI), normalized pigment chlorophyll ratio index (NPCI), and water band index (WBI), had strong correlations with HTI and with heat avoidance traits. This study highlights the contribution of pigments and wax as heat avoidance traits in crop canopies, and the potential application of spectral measurements for selecting genotypes with more heat resistant vegetation.

**Keywords:** pea; heat-stress; wax; lamina; petiole; canopy type; chlorophyll; anthocyanin; vegetation indices

## 1. Introduction

Pea (*Pisum sativum* L.) is a pulse crop that is widely grown in temperate regions for its nutritious seed and soil fertility benefits [1]. Unfortunately, pea is susceptible to heat stress, which causes impaired photosynthesis, accelerated senescence, and abortion of reproductive organs including flowers and pods, all culminating in reduced yield [2–4]. Due to weather alterations such as increased air temperature and severe drought caused by climate change, crop production is becoming increasingly challenging in many parts of the world [5]. For example, in the province of Saskatchewan, Canada, the worlds' leading producer and exporter of pea, the 2021 cropping season was the most heat and drought stressed in the last several decades, causing about 37% reduction in pea seed yield compared to the average of the previous five years (https://agriculture.canada.ca (accessed on 25 January 2022)). Pea heat stress arises in spring and summer-grown crops on days when air temperature exceeds a threshold of 28 °C, and when heat shock occurs from temperature > 34 °C for several hours during sensitive stages [3,6]. Although the extent

of heat sensitivity varies with phenology, heat stress can impede crop performance at any developmental stage [2,4].

To cope with heat and other sub-optimal environmental factors, plants have developed various amendments to their morpho-anatomical form and physiological and biochemical functions, as avoidance or tolerance strategies [7,8]. These strategies can be broadly categorized into long-term alterations to morphological architecture and phenological patterns, or short-term heat aversion mechanisms such as through transpirational cooling and reflection of radiation overload on plant canopies [9–11]. For example, spectral reflectance in the ultraviolet (UV) and infrared regions makes plants avoid or minimize radiation and heat load [10,12,13]. Such reflectance of excess heat can be affected by the amount and composition of epicuticular waxes [14]. Vegetation indices (VI), derived from spectral data, are useful proxies to qualitatively or quantitatively estimate traits associated with growth, biomass, pigment composition, and water content in a single leaf and at the canopy level in plant populations [15,16]. A recent study on wheat revealed the use of spectral data in predicting leaf epicuticular wax concentration [17].

Epicuticular wax making an outermost layer over plant surfaces protects the plant from extreme weather variables and contributes to the plant's survival under stressful environments [18]. In pea, epicuticular wax reduces residual transpiration, minimizing water loss to help maintain tissue water status under drought stress [19]. Likewise, pigments may be involved in heat tolerance through heat dissipation and protection of essential plant processes [9,10]. Recently, Arafa et al. [20] reported pea seed priming with carrot extracts rich in carotenoids enhanced the plant's biochemical functions, and contributed to greater yield and stress tolerance. Stay-green, a trait characterized by delayed plant senescence, contributes to improved yield under both drought and heat stress conditions [21].

Although selection for thicker leaf epicuticular wax as a drought tolerance trait has resulted in improved cultivars in several crops [19,22–24], its contribution to heat tolerance is usually overlooked. Similarly, leaf pigments and their association with heat tolerance or avoidance have not been sufficiently addressed. We hypothesized that increased leaf pigments and wax concentrations would contribute to pea heat stress avoidance, and a substantial range of concentration of these biochemical compounds would be distributed across diverse pea germplasm. Based on the association with leaf wax and pigments, vegetation indices may serve as proxies to indirectly determine plant's resistance to heat stress and, therefore, amenable to high throughput field phenotyping. Our specific objectives were: (1) to investigate heat stress effects on pea canopies varying in leaf pigment and wax concentrations, (2) to examine the contributions of leaf pigments and wax concentrations in heat avoidance in a diverse range of pea cultivars, and, (3) to determine how spectral vegetation indices associate with leaf pigments and wax concentrations.

## 2. Materials and Methods

### 2.1. Pea Germplasm and Growth Conditions

Twenty-four diverse pea cultivars, adapted to western Canada and described in Table 1, were tested in field trials for three years (2014–2016) at two locations, Rosthern (52°66′ N, 106°33′ W; Orthic Black Chernozem) and Saskatoon (52°12′ N, 106°63′ W; Dark Brown Chernozem), in Saskatchewan, Canada. The study consisted of six trial sets (environments): Rosthern 2014 (RL14), Saskatoon 2014 (SL14), Rosthern 2015 (RL15), Saskatoon 2015 (SL15), Saskatoon 2016 (SL16) and Saskatoon 2016 with a normal seeding date (SN16). All trial sets except SN16 were intentionally late seeded late by 20 to 30 days from the regular seeding date. Late seeding delayed reproduction and flowering duration into mid-July to early-August, where daytime maximum air temperatures rose to 27–35 °C for several days, imposing heat stress on pea.

**Table 1.** Canopy type, description, and name of 24 pea cultivars evaluated for heat resistance traits at Rosthern and Saskatoon, Canada, in 2014–16.

| Canopy Type [a] | Description | Cultivars [b] |
|---|---|---|
| 1-n-u-dg | normal leaf, upright habit, dark-green canopy | MPG87, MFR043, TMP 15213 |
| 2-n-v-bg | normal leaf, vining habit, bright-green canopy | Naparnyk, TMP 15116, TMP 15181, Torsdag, 40-10 |
| 3-n-v-dg | normal leaf, vining habit, dark-green canopy | Mini, Rally, Superscout |
| 4-sl-u-bg | semi-leafless, upright habit, bright-green canopy | Kaspa, CDC Sage, Aragorn, Eclipse |
| 5-sl-u-dg | semi-leafless, upright habit, dark-green canopy | 03H107P-04HO2026, 03H267-04HO2006, CDC Golden, CDC Vienna, CDC Meadow, Delta |
| 6-sl-v-dg | semileafless, vining habit, bright-green canopy | TMP 15179, TMP 15206 |
| 7-n-u-bg | normal leaf, upright habit, bright-green canopy | TMP 15202 |

[a] n, normal leaf; sl, semi-leafless; u, upright habit; v, vining habit; bg, bright-green color; dg, dark-green color.; [b] CDC, Crop Development Center; TMP, temporary accession designation.

The six environments were grouped into three environmental stress levels (ESL) based on temperature, vapor pressure deficit and precipitation: control (RL14 and SN16), intermediate (SL14 and SL16), and stress (RL15 and SL15). For every trial set of the six environments, a randomized complete block design was used with four replications. A standard plant breeding plot size, 1.37 m width × 3.66 m length with three raw seeding, was used. Errors associated with edge effects were minimized by bordering the plots by other pea plots. Plants were grown under best management practices recommended for pea production in western Canada. As details of the crop husbandry, including the type and active ingredients of herbicides used for weed control and fertilization, were described by Tafesse et al. [4], here we provide a brief summary of the practices. To control weeds, herbicides were applied in fall several months before seeding and during the trial seasons. The trials were seeded into cereal stubble, no fertilizer was applied, but seeds were inoculated with rhizobia for atmospheric nitrogen fixation. At maturity, a few days before harvesting, a desiccant was applied to facilitate uniform drying and the plots were combine harvested. Weather data for each of the six trial sets (environments) was presented by Tafesse et al. [4]. Generally, environments RL14 and SN16 had relatively cooler air temperature and sufficient rainfall, and were designated as control ESL. Environment SL14 and SL16 had relatively hotter conditions with sufficient rainfall and were considered as intermediate ESL, and environments RL15 and SL15 had the hottest and driest conditions compared to the other four environments and were designated as stress ESL.

*2.2. Leaf Sample Collection and Area Determination*

Pea plants have compound leaves consisting of stipules, leaflets, petioles, rachis, and tendrils [25]. The pea cultivars used in this study were normal and semi-leafless leaf types. Normal leafed cultivars had a wide flat leaf lamina surface from stipules, leaflets, and relatively short petioles and rachis. Semi-leafless cultivars had stipules, a longer petiole and rachis, with more tendrils, but no leaflets. Unless otherwise stated, we refer to the flat leaf surface as 'lamina', and the petiole, rachis plus tendrils as 'petiole'. Fully expanded young leaf samples from the second or third main stem node, counting down from the apical tip, were sampled for chlorophyll, carotenoid, anthocyanin and wax measurements. Three leaf samples were collected from each plot and placed in plastic bags in an ice box cooler to avoid evaporative loss prior to transport to the laboratory. The leaf samples were collected twice during the pea growing season, at early flowering, and the full seed stages. Leaf samples were clipped at the main stem node, separated into lamina and petiole components, and then lamina and petiole scanned separately for each plot using winRHIZO (Regent Instruments Inc, Quebec City, Canada) to determine their respective projected surface areas ($cm^2$). For each plot and time of collection, one leaf was used for chlorophyll and carotenoid extraction in acetone, another for anthocyanin in acidified ethanol, and the third leaf was used for wax extraction in chloroform.

### 2.3. Chlorophyll and Carotenoid Determination

Chlorophyll and carotenoid measurements were performed according to Lichtenthaler [26]. A 1.22 cm$^2$ stipule disc, and the entire petiole, using the same scanned tissue above, were each placed in 10 mL glass tubes with a tight cap, 3.5 mL of 100% acetone was added, and samples incubated for 6 h at room temperature for complete pigment extraction. Samples were then vortex mixed and centrifuged for 5 min at 5000 rpm. The supernatant was used for absorbance (A) measurement using an Agilent 8453 diode array spectrophotometer with 1.6 ± 0.5 nm resolution, equipped with Chem Station software for UV-visible spectroscopy (Agilent Technologies, Santa Clara, CA, USA), at wavelengths 470, 645, 662 and 710 nm. Concentrations of chlorophyll a, chlorophyll b, total chlorophyll, and total carotenoid were determined in µg cm$^{-2}$ using the following equations:

$$\text{Chlorophyll a (µg mL}^{-1}) = (11.24(A_{662} - A_{710})) - (2.04(A_{645} - A_{710})) \tag{1}$$

$$\text{Chlorophyll b (µg mL}^{-1}) = (20.13(A_{645} - A_{710})) - (4.19(A_{662} - A_{710})) \tag{2}$$

$$\text{Total Chlorophyll (µg mL}^{-1}) = (7.05(A_{662} - A_{710})) + (18.09(A_{645} - A_{710})) \tag{3}$$

$$\text{Total carotenoid (µg mL}^{-1}) = (1000(A_{470} - A_{710})) - (1.90 \text{ Chlorophyll a}) - (63.14 \text{ Chlorophyll b}) \tag{4}$$

In order to present the data in µg cm$^{-2}$, the results from the above equations were multiplied by the extraction volume (3.5 mL), and then divided by the sample (lamina or petiole) projected area (cm$^2$).

### 2.4. Anthocyanin Determination

A 1.33 cm$^2$ disc cut from stipule, and the entire petiole were each used per leaf, and anthocyanin extraction was done according to Abdel-Aal and Hucl [27]. Samples were placed in 5 mL glass tubes with 3 mL of acidified ethanol (85:15, V/V) ratio using 95% ethanol and 1.428N HCl) at pH 1, tubes were capped and then incubated overnight at room temperature for extraction. Samples were vortex mixed and centrifuged at 5000 rpm for 5 min. The supernatant solution was measured with spectrophotometer and absorbance was read at 535 and 663 nm, wavelength peaks for absorbance of anthocyanin and chlorophyll a, respectively. Total anthocyanin concentration in µg cm$^{-2}$ was calculated according to Murray and Hackett [28] as:

$$\text{Anthocyanin (µg ml}^{-1}) = A_{535} - 0.24(A_{663}) \tag{5}$$

### 2.5. Bulk Wax Determination

Leaf lamina and petiole wax were extracted and quantified according to the methods used on pea [19]. Details of the method used for bulk wax extraction and quantification is exactly as presented by Tafesse et al. [29]. The method can be summarized as follows: first, bulk was extracted from the leaf surfaces by dipping the leaf tissue (lamina or petiole) samples into 10 mL chloroform for 15 s in 100 mL glass tubes. To evaporate the chloroform, the tubes were then placed in a water bath at 70 °C for 30 min. Then 5 mL reagent (acidic potassium dichromate) was added to each tube containing the wax, and boiled at 100 °C for half an hour. After cooling, 5 mL distilled water was added to each tube, vortexed, and spectral absorbance was measured at 590 nm with a spectrophotometer. Finally, wax concentrations were calculated from the spectral data using a standard curve equation that was developed from a linear (R$^2$ > 0.98) relationship of known concentrations of beeswax [29].

### 2.6. Spectral Reflectance and Vegetation Indices

Spectral reflectance measurements on stipules were taken on three to five occasions per plot for each of the six environments during the crop reproductive phase using a portable spectroradiometer (Model PSR-1100F, Spectral Evolution Inc., Lawrence, MA, USA). This instrument enabled hyperspectral readings with a range of 320–1126 nm and

1.6 nm sampling interval, and a total of 512 discrete narrow bands. A 1-m fiber-optic cable with industry-standard interface with the instrument, controlled by a PSR-1100 Pistol Grip, enabled us to specifically capture reflectance from stipules for spectral measurements. A stipule of a fully expanded leaf at the second or third node counting from the tip of the pea main stem, fully exposed to the sun, was measured on sunny and usually hot days around solar noon (between 11:00 and 14:00 h) from the same direction, avoiding shadows, cloud, and any other interference we could control. Before measurements, reflectance was taken on a white plate that provided maximum reflection, and leaf reflectance was measured by holding the fiber sensor within 3 cm from the stipule surface approximately within a viewing angle of 80–90°. The reference reflectance was repeatedly taken every 15 min (equivalent to once every 12 plots) to adjust for the changing irradiance from the sun, and more frequently if clouds stopped measurements.

Vegetation and pigment indices, including normalized difference vegetation index (NDVI), green normalized difference vegetation index (GNDVI), photochemical reflectance index (PRI), normalized pigment chlorophyll ratio index (NPCI), and water band index (WBI), were each calculated according to Rouse et al. [30], Gitelson et al. [31], Gamon et al. [32], Peñuelas et al. [33], and Peñuelas et al. [34], respectively, as follows:

$$\text{NDVI} = (R_{nir} - R_r) \div (R_{nir} + R_r) \tag{6}$$

$$\text{GNDVI} = (R_{nir} - R_g) \div (R_{nir} + R_g) \tag{7}$$

$$\text{PRI} = (R_{531} - R_{570}) \div (R_{531} + R_{570}) \tag{8}$$

$$\text{NPCI} = (R_{531} - R_{570}) \div (R_{531} + R_{570}) \tag{9}$$

$$\text{WBI} = R_{900} \div R_{970} \tag{10}$$

where; R, reflectance; nir, near infrared band (bandwidth 760–860, center band 820 nm), r, red band (bandwidth 650–700 nm, center band 675 nm); g, green (bandwidth 530–580, center band 555 nm). The center bands were rounded to the nearest whole number (for example 530.5 nm was 531 nm). For vegetation indices calculated from two or more single bands such as WBI, the nearest whole number band was used as the center band.

## 2.7. Canopy Temperature

Canopy temperature was measured five to eight times per plot in each environment during late vegetative and reproductive growth using a handheld infrared (IR) thermometer (Model 6110.4ZL, Everest Interscience Inc., Tucson, AZ, USA). Measurements were taken within 3 h centred on solar noon when pea transpiration was at its maximum rate, assuming no drought stress-related closure, with the sun unobstructed by cloud, and when there was low wind pressure. The infrared thermometer was held for six seconds approximately 30 cm above the canopy at 15° field of view pointing down for a wider canopy view. During the six seconds of viewing the canopy, the thermometer averaged over a range of measurements and stabilized to the mean value that was used as one data point to represent a plot reading. The reading did not include any ground or soil surface, only green vegetation, and predominantly upright vegetation and the upper half of the canopy.

## 2.8. Heat Tolerance Index

Heat tolerance index (HTI), a concept that indicates the extent of yield reduction due to heat stress compared to the potential yield under control condition, was determined according to Fernandez [35] and used to separate cultivars yield response into heat sensitive and heat tolerant. The seed yield was obtained by small plot combine harvest of individual plots at maturity.

$$\text{HTI} = \frac{(\text{Yield}_c)\,(\text{Yield}_h)}{(\text{Yield}_{c.ave})^2} \tag{11}$$

where Yield$_c$ is seed yield for each cultivar in a replication under non-heat stress (control conditions), Yield$_h$ is seed yield for each cultivar in a replication under heat stress, Yield$_{c.ave}$ is the grand mean seed yield from all control plots of all replications per environment under non-heat stress conditions. When HTI is close to zero or zero, crops do not yield under heat and are heat sensitive. When HTI is high (>1), then the cultivar would be deemed heat tolerant for yield compared to the grand mean yield under the control conditions.

*2.9. Data Analysis*

Univariate analysis of the variables chlorophyll a, chlorophyll b, chlorophyll a/b, total carotenoid, total anthocyanin, and total wax concentrations from lamina and petiole, NDVI, GNDVI, PRI, NPCI and WBI were computed by using the mixed procedure of SAS, Version 9.4, SAS Institute. Before undertaking the analysis variance (ANOVA), normal distribution of residuals and homogeneity of variances, the two major ANOVA assumptions, were checked according to Shapiro-Wilk and Levene and tests, respectively [36,37] and these assumptions were met for each variable. Then ANOVA with the least square difference (LSD) test ($p < 0.05$) was performed on each variable. The designation of environments into three stress levels, "control", "intermediate" and "stress", was based on the intensity of stress at each environment as described by Tafesse et al. [4]. For growth habit, leaf type, and canopy color, seven combinations of canopy groups, designated as "type" were used for testing the effect of canopy type. Thus, the three main treatment factors were environmental stress that we simply referred to as 'environmental stress level (ESL)', canopy 'type', and 'cultivar'. The effects of ESL, type, cultivar, ESL by cultivar, and ESL by type nested in cultivar interactions were treated as fixed effects, and block nested in environment was treated as a random effect. Whenever the interaction term was significant, a separate analysis was performed for each of the three ESLs and the results of the 'control' and 'stress' levels are shown while the result of the 'intermediate' that generally lay between the two ESLs is omitted in figures to save space. Pearson correlations test were performed among the variables of canopy temperature, pigments, wax, and vegetation indices, and significance was declared at $p < 0.05$ for combined data using four environments, control and stress.

**3. Results**

*3.1. Treatment Effects on Pigment, Wax and Vegetation Indices*

The main treatment effects of ESL, canopy type and cultivar significantly affected all pigment and wax traits. The one exception was petiole anthocyanin which was not significantly affected by cultivar. Vegetation indices including PRI, GNDVI, WBI, and NPCI were significantly affected by all main effect treatment factors. The interaction of ESL by type impacted all traits except petiole anthocyanin and the majority of the vegetation indices. Only PRI, WBI and NPCI were affected by the interaction of ESL by type. The ESL by cultivar interaction affected lamina chlorophyll a, lamina and petiole chlorophyll b, lamina carotenoid, lamina and petiole wax concentration, GNDVI, WBI and NPCI, but not other traits (Table 2).

For ESL, compared to the control, stress decreased mean lamina chlorophyll a and chlorophyll b concentrations by 22.8, and 34.9%, respectively. In contrast, stress increased the corresponding chlorophyll a/b ratio, anthocyanin, and bulk wax concentrations by 23.9, 24.5, and 28.4%, respectively (Table 2). The increased chlorophyll a/b ratio under stress resulted from a greater reduction in chlorophyll b concentration than in chlorophyll a (Table 2; Figure 1A,B).

**Table 2.** Univariate analysis of variance (ANOVA) results showing significance of cultivar, environmental stress level (ESL), canopy type (T) main effects and their interactions on leaf pigments, wax, and vegetation indices of 24 pea cultivars grown across six environments at Rosthern and Saskatoon, 2014–16. Means of traits are presented for the three stress levels of control, intermediate, and stress. The control ESL are 2014 late seeding date at Rosthern and 2016 normal seeding date at Saskatoon; intermediate ESL are 2014 and 2016 late seeding date at Saskatoon; and stress ESL are 2015 late seeding date at Rosthern and Saskatoon, Canada.

| Variable | Tissue | ESL | T | Cultivar | ESL * T(C) | ESL * C | ESL Treatment Means Control | Intermediate | Stress |
|---|---|---|---|---|---|---|---|---|---|
| Chlorophyll a ($\mu$g cm$^{-2}$) | Lamina | *** | *** | *** | ** | ** | 33.5 a | 28.5 b | 25.9 c |
| | Petiole | * | *** | *** | * | NS | 20.9 a | 18.5 ab | 16.9 b |
| Chlorophyll b ($\mu$g cm$^{-2}$) | Lamina | *** | *** | *** | ** | ** | 11.0 a | 8.3 b | 7.2 c |
| | Petiole | * | *** | *** | ** | * | 9.2 a | 6.7 b | 6.0 b |
| Chlorophyll a/b ratio | Lamina | *** | *** | ** | *** | NS | 3.1 c | 3.57 b | 3.9 a |
| | Petiole | ** | *** | *** | ** | NS | 2.5 b | 2.93 a | 2.6 ab |
| Carotenoids ($\mu$g cm$^{-2}$) | Lamina | ** | *** | *** | * | * | 8.4 a | 7.83 a | 6.79 b |
| | Petiole | *** | *** | *** | ** | NS | 5.8 a | 4.98 b | 4.7 c |
| Anthocyanins ($\mu$g cm$^{-2}$) | Lamina | * | *** | *** | *** | NS | 1.06 b | 1.10 b | 1.32 a |
| | Petiole | * | *** | NS | NS | NS | 1.34 ab | 1.29 b | 1.37 a |
| Bulk wax ($\mu$g cm$^{-2}$) | Lamina | ** | *** | *** | *** | ** | 23.6 b | 27.34 a | 30.3 a |
| | Petiole | ** | *** | *** | ** | * | 41.6 b | 42.45 b | 53.4 a |
| NDVI | Lamina | ** | NS | * | NS | NS | 0.81 a | 0.77 b | 0.76 b |
| PRI | Lamina | ** | *** | *** | * | NS | −0.01 a | −0.02 b | −0.02 b |
| GNDVI | Lamina | ** | *** | *** | NS | * | 0.64 a | 0.61 b | 0.60 b |
| WBI | Lamina | *** | *** | *** | * | * | 1.10 a | 1.08 b | 1.08 b |
| NPCI | Lamina | ** | *** | *** | *** | * | 0.29 c | 0.33 b | 0.36 a |

*, **, and *** indicates significant at 0.05, 0.01, and 0.001 probability levels, respectively. NS, not significantly different at 0.05 probability level. For each variable within each raw, means labeled by same letter are not significantly different at 0.05 probability level. For each stress level, N = 192 from 24 cultivars, two environments and four replications. GNDVI, green normalized difference vegetation index; NDVI, normalized difference vegetation index; NPCI, normalized pigment chlorophyll ratio index.; PRI, photochemical reflectance index; WBI, water band index.

As reproduction proceeded from early flowering to pod filling, the leaf lamina chlorophyll a, chlorophyll b, carotenoid, and wax concentrations increased by 20, 18, 5, and 39%, respectively, and the corresponding anthocyanin concentration decreased by 20%. Similarly, petiole chlorophyll a, chlorophyll b, carotenoid, and wax concentrations increased by 28, 10, 6, and 53%, respectively (Figure 2). Generally, leaf lamina had >32% greater chlorophyll and carotenoid concentrations than those found in the petiole. On the other hand, higher anthocyanin and wax concentrations were found in the petioles compared with the leaf lamina (Table 2). Under both control and stress conditions, cultivars with dark green canopies, including Superscout, Rally, MPG87, Mini, and CDC Vienna, were associated with greater lamina chlorophyll and carotenoid concentrations. In contrast, the bright green cultivars including TMP15116, Naparnyk, CDC Sage, and Torsdag had less (<32 $\mu$g cm$^{-2}$) lamina chlorophyll and carotenoid concentrations (Figure 1A,B,D).

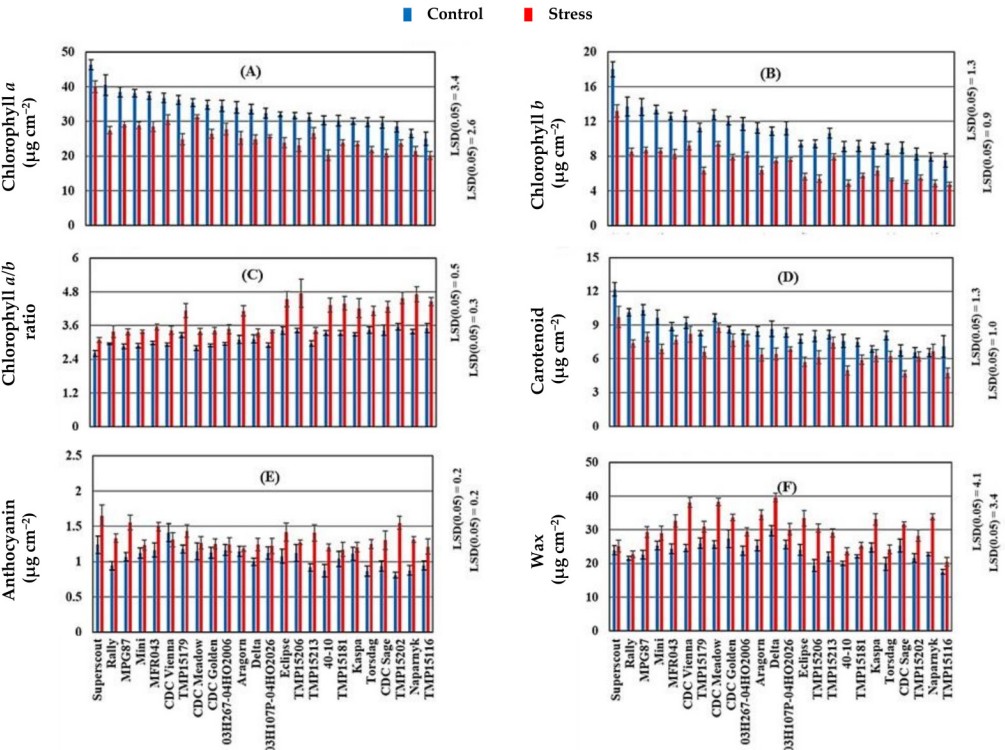

**Figure 1.** Mean lamina pigment and wax concentrations of 24 pea cultivars grown in control (blue) and stress (red) conditions, with (**A**) chlorophyll a, (**B**) chlorophyll b, (**C**) chlorophyll a/b ratio, (**D**) carotenoid, (**E**) anthocyanin and (**F**) wax concentrations ($\mu g \, cm^{-2}$). Each bar represents the mean values, and error bars are the standard error of the mean. For each bar, N = 16 (two environments × four replications × two growth stage samples) for each of control or stress condition. The LSD values for each of the stress levels is shown in the figure. The control conditions are 2014 late seeding date at Rosthern and 2016 normal seeding date at Saskatoon; and the stress conditions are 2015 late seeding date at Rosthern and Saskatoon, Canada.

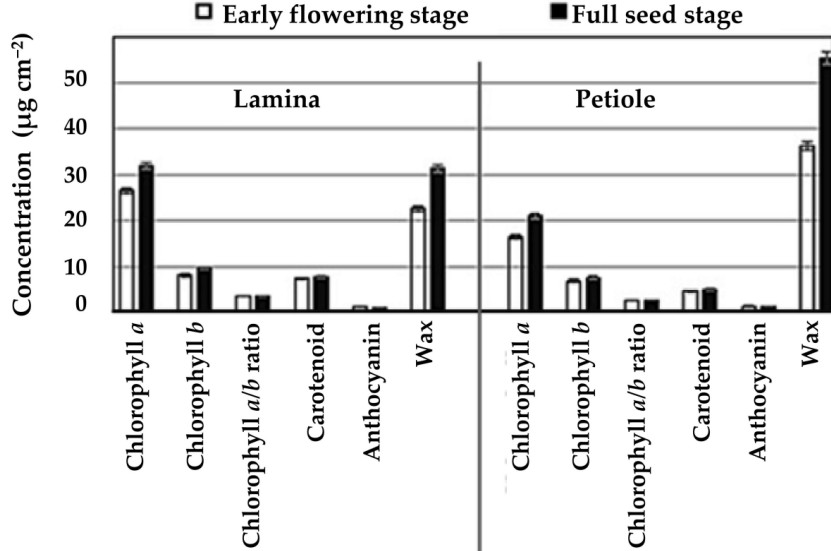

**Figure 2.** Chlorophyll a, b, a/b, carotenoid, anthocyanin and wax concentrations ($\mu g \, cm^{-2}$ of tissue) in leaf lamina and petiole at early flowering and flower termination stages of pea grown in six field environments in Saskatchewan, Canada (2014 to 2016). Each bar is the mean value averaged over 24 cultivars, six environments and four replications per environment, i.e., N = 576 for each variable.

Generally, for chlorophyll and carotenoid concentrations under both control and heat stress, the normal leafed vining cultivars with dark green canopies had the greatest chlorophyll a, chlorophyll b, and carotenoid concentrations; but the least chlorophyll and carotenoid concentrations were found in normal leafed vining cultivars with bright green color (Figure 3A,B,D). Under control and heat stress, bright green cultivars had a higher chlorophyll a/b ratio than dark green cultivars regardless of the growth habit and leaf type (Figure 3C). Under control conditions, normal leafed vining cultivars with bright green canopies had lower anthocyanin concentration than all other types; but under heat stress this type had a relatively greater anthocyanin concentration than other types (Figure 3E). For leaf wax, under control conditions, semi-leafless cultivars had the same wax concentration regardless of their canopy habit and color. The lowest wax concentrations were found in normal leafed vining cultivars with bright green canopies (Figure 3F). Under heat stress, upright semi-leafless cultivars with dark green canopies had the greatest wax concentrations, and normal leafed vining cultivars had the lowest wax concentrations (Figure 3F).

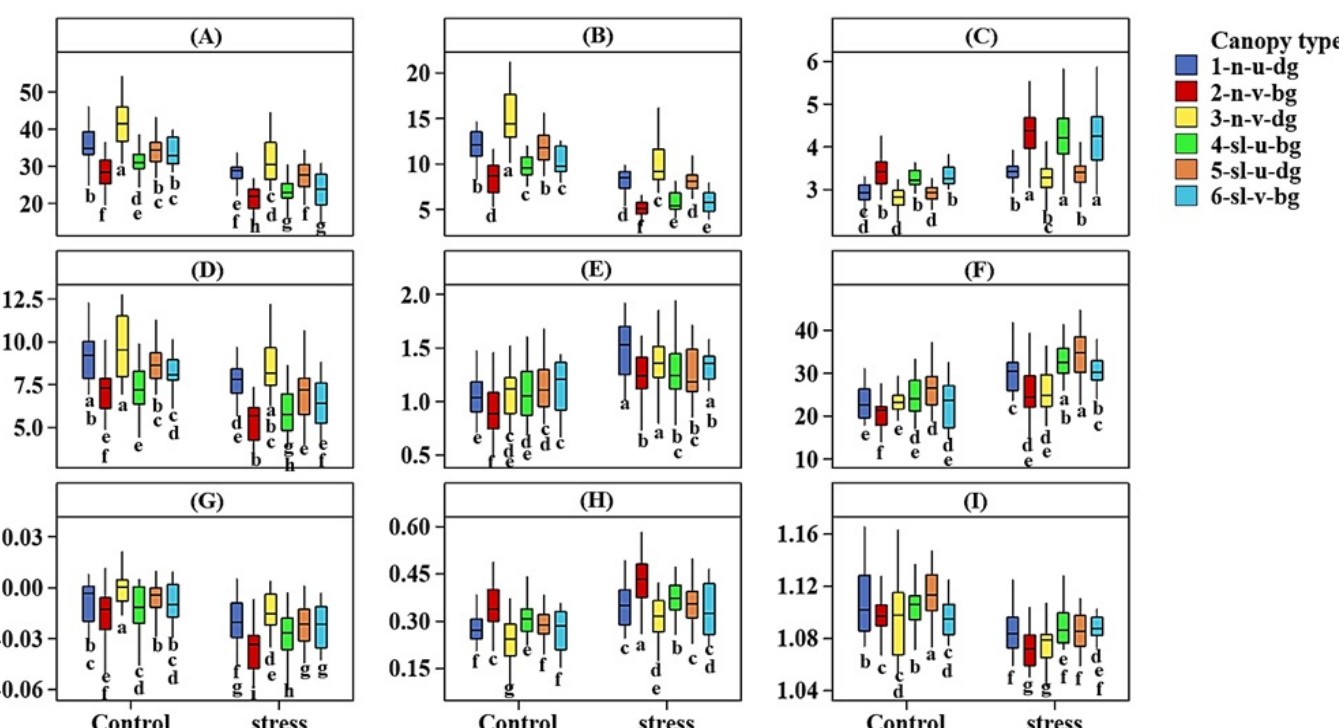

**Figure 3.** Box plot showing the interaction effects of heat stress and canopy type on: chlorophyll a (**A**), chlorophyll b (**B**), chlorophyll a/b ratio (**C**), carotenoid (**D**), anthocyanin (**E**), wax (**F**) concentrations ($\mu g$ $cm^{-2}$); and vegetation indices: PRI (**G**), NPCI (**H**), and WBI (**I**) measured on pea leaf stipules under control and stress environments at Rosthern and Saskatoon, Canada, 2014–16. The size of box represents 50% of the middle data, the line in the middle of the box is the median, and the whiskers represent the range of the data. Boxes labeled with same letters within trait are not significantly different at $p < 0.05$. N = 24 for type 1, 40 for type 2, 24 for type 3, 32 for type 4, 48 for type 5, and 16 for type 6. The control conditions are 2014 late seeding date at Rosthern and 2016 normal seeding date at Saskatoon; and the stress conditions are 2015 late seeding date at Rosthern and Saskatoon, Canada.; Canopy type legend: n, sl, u, v, bg, and dg represents normal leaf, semi-leafless; upright habit, vining habit, bright-green color, and dark-green color, respectively; and vegetation indices: PRI, photochemical reflectance index; NPCI, normalized pigment chlorophyll ratio index; WBI, water band index.

### 3.2. Response of Vegetation Indices

The control environment had greater NDVI and GNDVI values than the stress and intermediate environments. Although the values of most vegetation indices were in the 'normal' range for healthy vegetation, heat stress and control treatments differed (Table 2). For PRI, under both control and heat stress, dark green cultivars had greater PRI than bright green cultivars regardless of leaf type and canopy habit (Figure 3G). Under both control and stress conditions, normal leafed vining cultivars with bright green canopies had greater NCPI, suggesting more stress than all other canopy combinations (Figure 3H). For WBI, for the control, semi-leafless upright cultivars with dark green canopies had a greater WBI than vining cultivars regardless of leaf type and canopy color. Furthermore, under heat stress, semi-leafless upright cultivars with dark green canopies had the greatest WBI, inferring a high leaf water content compared to all other canopy types. For WBI and heat stress, the cultivar ranking matched with the cultivar ranking for wax concentration. Water band index is associated with leaf water content, so the greater WBI value in upright and semi-leafless cultivars implied that these cultivars maintained greater leaf water content under heat stress.

### 3.3. Heat Tolerance Index

Pea cultivars significantly varied in HTI, calculated from the relative seed yield of cultivars under control and heat stress, with values ranging from 0.35 to 1.25 (Figure 4A). Generally, upright cultivars with dark green canopies had a greater HTI, with the smallest HTI in normal leafed vining cultivars with dark green canopies (Figure 4B). Cultivars with greater (>1) HTI included CDC Meadow, TMP 15 2013, CDC Golden, Naparynk and TMP 15181 (Figure 4A). Heat tolerance index was negatively correlated with canopy temperature (Figure 5C), positively correlated with lamina wax concentration and WBI (Figure 5E,F).

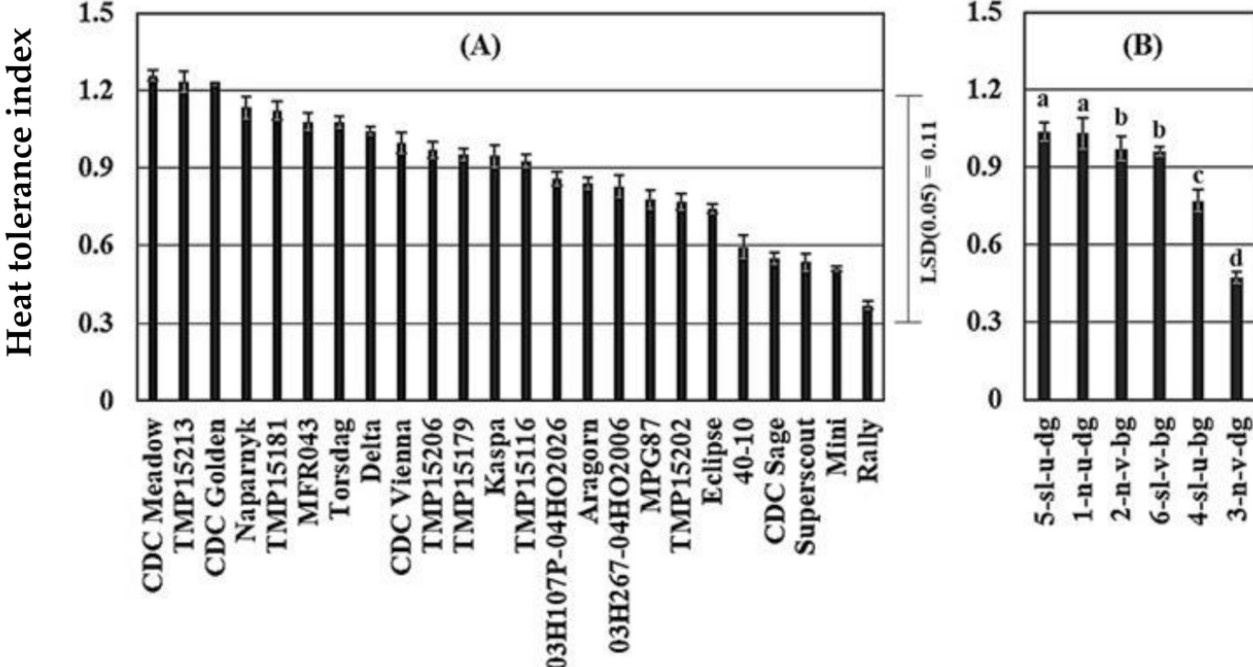

**Figure 4.** Heat tolerance index by cultivars (**A**), and canopy type (**B**) of 24 pea cultivars grown in four field environments (control and stress) in western Canada. N = 8 for each cultivar in panel A; and N = 12 for type 1, 20 for type 2, 12 for type 3, 16 for type 4, 24 for type 5, and 8 for canopy type 6 in panel B. Error bars are standard errors of the mean. In panel B, canopy types labeled with same letters are not significantly different at $p < 0.05$. Legend for canopy type: n, normal leaf; sl, semi-leafless; u, upright habit; v, vining habit; bg, bright-green color; dg, dark-green color.

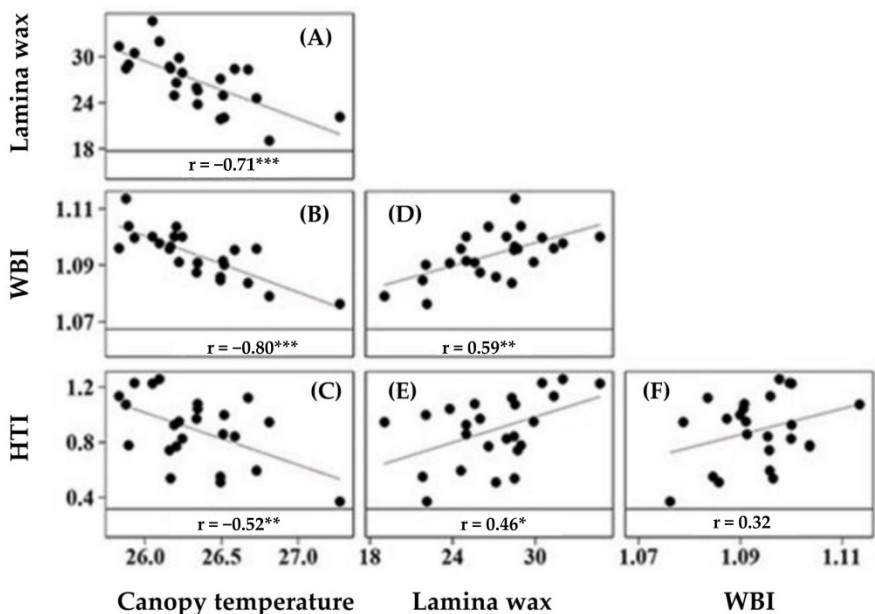

**Figure 5.** Matrix plot showing correlation between canopy temperature (CT) and lamina wax (**A**), CT and WBI (**B**), CT and HTI (**C**), Lamina wax and WBI (**D**), Lamina wax and HTI (**E**), and WBI and HTI (**F**) of 24 pea cultivars grown in four field environments (control and stress) at Rosthern and Saskatoon, western Canada, in 2014−16. Each symbol is a cultivar averaged over 16 observations (four environments by four replications). *, **, and *** indicates significant at 0.05, 0.01, and 0.001 probability levels, respectively. HTI, heat tolerance index; WBI, water band index.

*3.4. Phenotypic Correlation among Pigments, Wax, Vegetation Indices, Canopy Temperature, and Heat Tolerance Index*

　　　Leaf spectral reflectance is mainly affected by pigment and wax compositions and concentrations. Our result showed that most vegetation indices had significant correlation with pigment and wax concentrations, canopy temperature and HTI (Table 3 and Figure 5). The PRI and GNDVI had similar trends, correlating positively with chlorophyll, carotenoid, and anthocyanin concentrations, and negatively with chlorophyll a/b ratio. In contrast, NPCI had the opposite pattern to PRI and GNDVI. Correlations among lamina wax concentration, WBI, canopy temperature, and HTI were of specific interest and illustrated in Figure 5. Water band index correlated positively with lamina wax concentration, and negatively with canopy temperature. Finally, Heat tolerance index was negatively correlated with canopy temperature (Figure 5C), and positively correlated with lamina wax concentration and WBI (Figure 5E,F).

**Table 3.** Pearson correlation test showing associations among canopy temperature, total chlorophyll, chlorophyll a/b ratio, carotenoid, anthocyanin, wax, photochemical reflectance index (PRI), green normalized vegetation index (GNDVI), water band index (WBI), and normalized pigment chlorophyll ratio index (NPCI) of 24 pea cultivars grown under field conditions under control (upper right diagonal) and stress (lower left diagonal) environments averaged over two environmental levels and four replications. The control conditions are 2014 late seeding date at Rosthern and 2016 normal seeding date at Saskatoon; and the stress conditions are 2015 late seeding date at Rosthern and Saskatoon, Canada.

| Variable | Canopy Temperature | Total Chlorophyll | Chlorophyll a/b Ratio | Carotenoid | Antho-cyanin | Wax | PRI | GNDVI | WBI | NPCI |
|---|---|---|---|---|---|---|---|---|---|---|
| Canopy temperature | | 0.23 | −0.03 | 0.29 | −0.31 | −0.46 * | 0.06 | 0.07 | −0.55 * | −0.07 |
| Total chlorophyll | −0.16 | | −0.86 *** | 0.95 *** | 0.61 ** | 0.37 | 0.74 *** | 0.64 ** | 0.12 | −0.77 *** |
| Chlorophyll a/b ratio | 0.19 | −0.80 *** | | −0.85 *** | −0.60 ** | −0.46 * | −0.61 ** | −0.47 * | −0.22 | 0.55 ** |
| Carotenoid | −0.18 | 0.92 *** | −0.76 *** | | 0.53 ** | 0.23 | 0.68 ** | 0.62 * | −0.01 | −0.66 ** |
| Anthocyanin | −0.03 | 0.50 * | −0.20 | 0.47 * | | 0.45 * | 0.44 * | 0.37 | 0.25 | −0.43 * |
| Wax | −0.72 *** | 0.15 | −0.18 | 0.26 | −0.07 | | 0.34 | 0.17 | 0.50 * | −0.31 |
| PRI | −0.21 | 0.81 *** | −0.73 *** | 0.72 *** | 0.39 | 0.09 | | 0.67 ** | 0.21 | −0.81 *** |
| GNDVI | −0.13 | 0.78 *** | −0.65 ** | 0.64 ** | 0.55 ** | −0.05 | 0.74 *** | | 0.40 * | −0.79 *** |
| WBI | −0.67 ** | 0.12 | −0.05 | 0.13 | 0.01 | 0.52 ** | 0.24 | 0.15 | | −0.31 |
| NPCI | 0.10 | −0.74 *** | 0.61 ** | −0.64 ** | −0.48 * | −0.11 | −0.75 *** | −0.70 ** | −0.16 | |

*, **, and *** indicates significant level at the 0.05, 0.01, and 0.001 probability levels, respectively.

## 4. Discussion

### 4.1. Leaf Pigment Concentrations as Heat Resistance Traits

We found that heat stress and the significant cultivar by environment interaction lowered chlorophyll a and chlorophyll b concentrations in leaf lamina and petiole, with the reduction more pronounced in heat sensitive cultivars. Photosyntheric pigments are prone to heat and other environmental stresses. Recently, Giordano et al. [38] reviewed the reduction of photosynthetic pigments in response to heat stress, and such reduction led to the reduction of photosynthetic activities related to photosystem II. Cultivars that were able to maintain chlorophyll concentration under heat stress had greater HTI values, and therefore greater heat resistance, implying that chlorophyll concentration was likely linked to plant heat response. As chlorophyll is an integral component in light absorption and transfer; chlorophyll loss or degradation leads to reduced photosynthesis and coupled oxidative damage which consequently reduces growth and yield [7,11]. Under heat and excess radiation stress, chlorophyll loss arises either due to limited biosynthesis caused by enzyme malfunctioning [39], or due to rapid degradation caused by heat and radiation damage. Chlorophyll loss also occurs naturally in senescing plants, and stress induces tissue senescence [40].

Interestingly, chlorophyll a/b ratio increased under heat stress in both leaf lamina and petiole in our research, likely due to rapid chlorophyll b degradation compared to that of chlorophyll a, suggesting a differential susceptibility in light-harvesting chlorophyll a/b-binding protein complex [41]. Although chlorophyll a/b ratio changes were associated with plant heat response, the literature is inconsistent in how chlorophyll a/b ratio changes with stress in crops. Feng et al. [42] found decreased chlorophyll a/b ratio was associated with heat tolerance in wheat, but Cui et al. [43] reported the opposite on a cool season perennial grass tall fescue (*Festuca arundinacea*). While the optimal range of chlorophyll a/b ratio needs further study, we noted that pea cultivars with either high (>4.0), or low (<2.5) chlorophyll a/b ratio had low heat tolerance indices (Figures 1C and 5A), suggesting damage at the antenna complex or reaction center, respectively, as reported by Feng et al. [42]. We found, generally, that upright cultivars with dark green canopies had low chlorophyll a/b ratio, and greater HTI than vining cultivars with bright green canopies (Figures 2C and 4B), inferring that upright canopies were less stressed.

Leaf lamina carotenoid concentration had a similar pattern as chlorophyll concentration and decreased due to the heat stress (Figures 1C and 3D). In published research, carotenoid biosynthesis and accumulation were influenced by multiple factors including light and temperature stresses [44]. Although heat stress resulted in a decreased concen-

tration of carotenoid, there was a significant difference among the pea cultivars. Cultivars better able to maintain relatively stable carotenoid concentration under heat stress had greater HTI (Figures 1D and 4A), implying that greater or maintained carotenoid concentration reduced heat damage on pea seed yield. Carotenoids are antenna pigments and have direct influence on photosynthesis, their two major roles being light harvesting during photosynthesis, and minimizing photo-oxidative damage of chlorophyll molecules by dissipating excess energy in the form of heat [10,45] by the Mehler-ascorbate-peroxidase cycle at Photosystem I [46].

Anthocyanin concentration increased with heat-stress (Table 2; Figure 1E), a pattern opposite to chlorophyll and carotenoid concentrations. Anthocyanin production is enhanced in response to most environmental stresses including cold, heat, drought, and light [47]. However, stressful environments also trigger formation of reactive oxygen species and free radicals [48]. To protect plants from the damaging effects of reactive oxygen species, high levels of anti-oxidants are required, and anthocyanins fulfill such a protective role [47]. Anthocyanins protect chloroplasts by reducing incident light, and they have an anti-oxidant role through scavenging reactive oxygen species [49]. Unlike chlorophyll and carotenoid, anthocyanin concentration was greater in petiole than lamina, and anthocyanin concentration declined in leaf lamina during reproduction, indicating anthocyanin biosynthesis was growth-stage dependent and younger leaves produced more. Anthocyanins also function like sunscreen for leaves, where anthocyanins form a layer and damaging radiation does not penetrate internal sensitive tissue. In addition to heat and UV protection, increased anthocyanin concentration under heat stress is associated with enhanced water uptake and decreased transpiration [7].

### 4.2. Wax as Heat Resistance Trait

While the roles of epicuticular wax as a drought tolerance trait have been extensively reported in cereals and brassica crops [19,22,23,50], a heat avoidance role for wax has rarely been addressed. We found significant variation, ranging from 23 to 53 $\mu$g cm$^{-2}$, in lamina and petiole bulk wax concentrations under heat and control conditions. Wax composition and concentration shows variation within and across crop species [19,22,23]. Our results showed that compared to the control, heat-stress resulted in a 28% increase in total leaf wax concentration. Moreover, during reproduction, from early flowering to full seed stage, wax concentration increased by >45% in heat and control environments (Figure 2). Part of this wax increase can be due to a reduction in leaf expansion during the season as crops experience diminished water supply, and part of this increase is likely due to increased induction of leaf wax biosynthesis. Overall, our results indicate that genetic factors (cultivar), plant age and heat stress jointly contributed to effects on leaf wax biosynthesis. In addition to heat stress, various stresses such as drought, cold, salinity, and mechanical damage have each contributed to increased wax load in crops [19,23].

For heat avoidance, epicuticular wax has two major roles. First, guarding leaves and stems from radiation and heat loads by reflecting ultraviolet, visible and infrared wavelengths. In a pilot study in which extra wax was applied to pea leaf surfaces under field conditions, we recorded radiation reflectance in the visible and near-infrared region and found reflectance here was positively associated with wax concentration [51]. Second, by minimizing water loss through reduced stomatal and residual (i.e., non-stomatal) transpiration, several groups associated epicuticular wax with improved drought tolerance [22,52,53]. Drought and heat stress usually occur together, and drought stress aggravates plant heat stress. Heat stress can be moderated if the plant is able to maintain and conserve sufficient water in leaves and tissues for transpirational cooling while minimizing non-transpirational losses. Our results showed that greater wax concentration was generally associated with a cooler canopy temperature, and a higher heat tolerance index (Figure 5A,E).

We discovered that upright canopies have an advantage in stress, an important finding in pea where leaf type determines the fate of upright crops to stay upright or lodge and suffer high temperatures early in vegetative growth. Upright canopies have also been linked

to lower canopy temperatures versus lodged canopies in wheat [54]. Our pea cultivars with upright growth habits and semi-leafless leaf type, both stress hardy traits, were also associated with higher wax concentration under heat stress (Figure 3F). Wax accumulation was positively associated with WBI in both control and heat stress conditions; WBI is a proxy for leaf water content, indicating that leaf surface wax minimized water loss (Table 3). Thus, leaf wax indirectly functioned as a heat tolerance trait because sufficient water supply was able to moderate heat stress by 2 °C [51]. Similarly, Camarillo-Castillo et al. [17] reported the importance of leaf epicuticular wax in enhancing light reflecting both in the visible and near infrared regions, which likely contribute to the dissipation of heat and excess energy [52]. Generally, glaucousness or waxy leaves were associated with high water potential that contributed to cooler canopy [19]. We concluded that greater lamina and petiole wax concentrations minimized heat stress by guarding pea from excess radiation and heat, and they also helped maintain leaf water content by lowering residual transpiration.

*4.3. Spectral Reflectance Association with Heat Resistance Traits*

Recent advancements in large scale and more accurate phenotyping techniques largely rely on remotely sensed data. These measurements focus on leaf and canopy traits including vegetation area, pigments, canopy temperature and plant water status; all these being associated with a crops's overall physiological status [34,55]. For example, indices derived from reflectance in the visible and near infrared regions such as NDVI and its derivatives indicate vigor and biomass, vegetation greenness, photosynthesis efficiency, and rate of senescence [56,57]. In soybean, Dhanapal et al. [58] demonstrated useful correlations between leaf and canopy measured pigments and canopy measured VIs that are applicable for high throughput field phenotyping. A dark green canopy index was able to distinguish dark green genotypes from regular soybean genotypes, and showed several steps in nitrogen metabolism and transport, photosynthesis and senescence across a range of germplasm and environments [59]. Sexton et al. [60] reported that photosynthetic capacity of plants can be effectively determined non-destructively from hyperspectral reflectance in the short-wave infrared regions. More recently, Camarillo-Castillo et al. [17] showed the application of spectral data to assess epicuticular wax concentration in wheat leaves and the benefits of such information to indirectly select stress resistant wheat cultivars. Their study also showed the ability of spectral measurement to effectively predict leaf epicuticular wax concentration. Several single nucleotide polymorphism markers and candidate gene associated vegetation indices including NDVI, PRI, NPCI, and WBI were reported from recent studies conducted on pea [29,61]. These studies clearly indicated the importance of spectral data and vegetation indices in detecting plant stress responses and the associated genetic factors controlling such responses. We found that both genotype and environment had significant effect on pea pigment, and also on wax concentrations. Alteration of pigment and wax concentrations under various environments suggest direct involvement in avoiding or tolerating stress. Reflectance in the visible wavelengths (400–700 nm) are influenced mainly by leaf chlorophyll, carotenoid and anthocyanin concentrations and compositions [15,32,34].

Heat stress degrades photosynthetic pigments, and hampers the photosynthesis processes at different levels, and such effects can be indirectly traced from spectral reflectance. Our results demonstrated a positive correlation between GNDVI and chlorophyll concentration (Table 3). Vegetation indices derived from reflectance in the near infrared region including WBI are proxies for tissue water status [34]. We found WBI was negatively correlated with canopy temperature (Figure 5B), and positively correlated with wax concentration (Figure 5D). Another group of VIs are those derived from the reflectance in the visible spectral region including PRI and NPCI, proxies for pigment concentration and function, and photosynthesis [32,33]. Significant positive correlation was observed between PRI and chlorophyll concentration, and NPCI was associated with limited pigment and high stress. Such strong and consistent association of VIs with pigment, wax, canopy temperature and other stress related traits indicate the potential of the VIs specifically

GNDVI, PRI, NPCI and WBI, as measurement proxies in heat stress studies for pea and other crops.

## 5. Conclusions

Our results on pea demonstrated several novel findings. Firstly, heat stress reduced chlorophyll a, chlorophyll b, and carotenoid concentrations, but increased wax and anthocyanin concentrations, and chlorophyll a/b ratio in leaves. Generally, leaf pigments (chlorophyll, carotenoid, and anthocyanin) both from petiole and lamina were positively correlated with heat tolerance index and contributed to lower canopy temperature. Secondly, surface wax contributed to heat resistance presumably by reflecting excess radiation and heat from the plant canopy; and by minimizing water loss through reduced stomatal and residual (i.e., non-stomatal) transpiration. Thirdly, cultivars with the semi-leafless leaf type, upright habit, and dark-green canopies were associated with high (>1) HTI under the heat stress environments, inferring that these traits conferred heat resistance and upright, dark green canopies result in more stress resistant crops. Finally, vegetation indices including GNDVI, PRI, NPCI, and WBI measured from stipules showed consistent relationships with pigment and wax concentrations and other heat tolerance traits, suggesting these indices can be useful proxies in future heat stress studies, and for high throughput phenotyping for heat stress resistance.

**Author Contributions:** Conceptualization, R.B. and T.D.W.; methodology, E.G.T. and R.B.; software, E.G.T. and R.B.; validation, R.B., T.D.W., S.S. and S.N.; formal analysis, E.G.T.; investigation, E.G.T. and R.B.; resources, R.B. and T.D.W.; data curation, E.G.T.; writing—original draft preparation, E.G.T.; writing—review and editing, E.G.T., R.B., T.D.W., S.S. and S.N.; visualization, E.G.T., T.D.W. and R.B.; supervision, R.B. and T.D.W.; project administration, R.B.; funding acquisition, R.B. All authors have read and agreed to the published version of the manuscript.

**Funding:** This work was supported by the Saskatchewan Agriculture Development Fund, Saskatchewan Pulse Crop Development Board, and Western Grains Research Foundation.

**Institutional Review Board Statement:** Not applicable.

**Informed Consent Statement:** Not applicable.

**Data Availability Statement:** This manuscript includes the essential data used either as tables or figures in the Section 3.

**Acknowledgments:** The authors thank B. Louie, J. Denis, Z. Wang, S. Ryu, and R. Xiang for their assistance in field measurements, and the Crop Development Centre, University of Saskatchewan Pulse Crop Breeding staff for seeding the trials and plot management.

**Conflicts of Interest:** The authors declare that they have no conflict of interest.

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
