# Peer review of "Leaf Pigments, Surface Wax and Spectral Vegetation Indices for Heat Stress Resistance in Pea"

_agronomy, doi:10.3390/agronomy12030739_

Round 1

Reviewer 1 Report

Keywords; author can’t add the keywords from title or abstract, need to improve.

Introduction: Author’s should address link the heat stress with the climate change which I think missing in introduction part.

Every time when author’s use the work Chlorophyll a or Chlorophyll b , a and b should in intalic version i.e., Chlorophyll a  and Chlorophyll b.

Table 2: Table legend is too lengthy need to improve it or brief explanation

Figure 1: need to improve the figure because in Axis title author mention the Lamina pigment and Wax concentrations but in legend mentioned each figure have different axis title. Each figure should have separate axis title and units.

Figure 2: Same for figure 2. Need to clear in the legend of figure or improve it with the figure. Confusing figure and legend

Author Response

We appreciate the careful reviews we received which have helped to improve the manuscript. Following are our detailed responses to the reviewers’ suggestions.

Reviewer 1

Keywords; author can’t add the keywords from title or abstract, need to improve.

Authors’ reply: The keywords are now modified to pea, heat-stress, lamina, petiole, canopy-type, chlorophyll, anthocyanin, vegetation-indices

Introduction: Author’s should address link the heat stress with the climate change which I think missing in introduction part.

Authors’ reply: The introduction has now included more information on how climate change and heat stress are related.

Every time when author’s use the work Chlorophyll a or Chlorophyll b, a and b should in intalic version i.e., Chlorophyll  and Chlorophyll b.

Authors’ reply: Thank you for noticing this and we have changed chlorophyll a into chlorophyll a, chlorophyll b into chlorophyll b, and chlorophyll a/b ratio into chlorophyll a/b ratio.

Table 2: Table legend is too lengthy need to improve it or brief explanation

Authors reply: We tried to make it more concise; we also believe that every table has to stand alone and give the full description of all information included in the table.

Figure 1: need to improve the figure because in Axis title author mention the Lamina pigment and Wax concentrations but in legend mentioned each figure have different axis title. Each figure should have separate axis title and units.

Authors’ reply: We have considered the comments and modified the axis labeling accordingly. We also have improved the resolution of the figures.

Figure 2: Same for figure 2. Need to clear in the legend of figure or improve it with the figure. Confusing figure and legend

Authors’ reply: We have considered the comments and modified the legends of the figure to make it simple and clear.

Reviewer 2 Report

The article concerns the interesting topic of investigating the role of leaf pigments, surface wax and spectral vegetation indices in heat avoidance in pea canopies. The authors emphasize the determination of correlation between some indices that is a novelty in their research. However, there are a few things that need to be corrected and clarified.

Line 31: Too many Keywords are the same as in the title.

The Introduction section seems too general to me. Newer references should be added. There are no references from the last two years. 

Lines 84, 95: Could the authors provide any information despite the references?

Lines 99-100: Why is 'b' (line 99) before 'a' (line 100)?

Line 117: Why were measurements performed in only one repetition for each plot?

LInes 235-237: What tests were used?

The figures are of low resolution.  

Why were the data obtained so long ago (2014−2016)? Did the authors try to repeat the experiment in later years?

The results should also be discussed against the background of the current literature data. Many of the references are very old. I think newer references should be added. 

At the end of section 5. Conclusions, the authors indicated the directions of further research. Have the authors not undertaken additional research on this topic since 2016? 

Author Response

Reviewer 2

The article concerns the interesting topic of investigating the role of leaf pigments, surface wax and spectral vegetation indices in heat avoidance in pea canopies. The authors emphasize the determination of correlation between some indices that is a novelty in their research. However, there are a few things that need to be corrected and clarified.

Line 31: Too many Keywords are the same as in the title.

Authors’ reply: The keywords are now modified to pea, heat-stress, lamina, petiole, canopy-type, chlorophyll, anthocyanin, vegetation-indices

The Introduction section seems too general to me. Newer references should be added. There are no references from the last two years.

Authors’ reply: We have modified the Introduction to make to more specific to the topic. We have also included some new references that we think are relevant to the topic. However, over the last two years there has been not much new information published which is directly related to the topic of our paper.

Lines 84, 95: Could the authors provide any information despite the references?

Authors’ reply: In addition to referencing our recent publication for management practices and weather information, we have now added more information to make the section stand alone and more complete for our readers. Thank you for the valuable suggestion.

Lines 99-100: Why is 'b' (line 99) before 'a' (line 100)?
Authors reply: We omitted the ‘a’ and ‘b’ to avoid any confusion and we carefully modified the associated information.

Line 117: Why were measurements performed in only one repetition for each plot?

Authors’ reply: For the destructive measurements of each of the pigments and wax variables (wax, anthocyanin, and chlorophyll), we had 48 repetitions per cultivar per leaf tissue (lamina and petiole). Yes, only one leaf was sampled per plot at a time, but there were four biological replications (plots), six environments, and two samplings per season (4 x 6 x 2 = 48 data points for each of the petiole and lamina per cultivar). As there are two environments for each of the three environmental stress level (ESL), and 24 cultivars, the total data points for each ESL was  4 replication x 2 ESL x 2 sampling time points x 24  cultivars =   384.  Thus, the sampling, measurements, and statistical analyses were extensive. 

Lines 235-237: What tests were used?

Authors’ reply: normal distribution of residuals was checked using Shapiro-Wilk tests, and Levene test was used to check the homogeneity of variances. The section was modified to include this information.

The figures are of low resolution.  

Authors’ reply: We have improved the resolution of the figures. 

Why were the data obtained so long ago (2014−2016)? Did the authors try to repeat the experiment in later years?

Authors’ reply: Although this same experiment was not repeated, follow-up studies are underway, particularly in detecting the genetic regions controlling these traits.

The results should also be discussed against the background of the current literature data. Many of the references are very old. I think newer references should be added. 

Authors’ reply: We have considered the comment, modified the discussion by including some newer literature from the last two years. However, we have to admit that over the last two years, there is not much information available directly related to our topic.

At the end of section 5. Conclusions, the authors indicated the directions of further research. Have the authors not undertaken additional research on this topic since 2016? 

Authors’ reply: There are several studies currently in progress in our labs that involve large scale phenotyping using remote sensing technologies and vegetation indices. Of course, the result of this study is an important input in using vegetation indices as proxies to quantitatively estimate variety performance in regard to growth, phenology, seed yield and response to various environmental stresses.  In addition, using some of the VI and other traits reported here, we have identified several single nucleotide polymorphisms and candidate genes associated with the VIs and related traits in genome-wide association studies (Tafesse et al., 2020, Tafesse et al., 2021), We also have undertaken a transcriptome and gene expression study and identified several genes related to pea heat stress response (Huang et al., 2021).  

Round 2

Reviewer 2 Report

The manuscript has been corrected according to all my comments